

# A machine learning-based model for clinical prediction of distal metastasis in chondrosarcoma: a multicenter, retrospective study

Jihu Wei[1,*], Shijin Lu[2,*], Wencai Liu[3], He Liu[1], Lin Feng[1], Yizi Tao[1], Zhanglin Pu[1], Qiang Liu[4], Zhaohui Hu[5], Haosheng Wang[6], Wenle Li[7], Wei Kang[8,9], Chengliang Yin[8] and Zhe Feng[10]

[1] Faculty of Postgraduate, Guangxi University of Chinese Medicine, Nanning, Guangxi, China
[2] Centre for Translational Medical Research in Integrative Chinese and Western Medicine, Ruikang Hospital Affiliated to Guangxi University of Chinese Medicine, Nanning, Guangxi, China
[3] Department of Orthopaedics, Shanghai Jiao Tong University Affiliated Sixth People's Hospital, Shanghai, China
[4] Orthopedic Department, Xianyang Central Hospital, Xianyang, Shannxi, China
[5] Department of Spine Surgery, Liuzhou People's Hospital, Liuzhou, Guangxi, China
[6] Department of Orthopaedics, The Second Hospital of Jilin University, Changchun, China
[7] State Key Laboratory of Molecular Vaccinology and Molecular Diagnostics & Center for Molecular Imaging and Translational Medicine, School of Public Health, Xiamen University, Xianmen, Fujian, China
[8] Faculty of Medicine, Macau University of Science and Technology, Macau, China
[9] Department of Mathematics, Physics and Interdisciplinary Studies, Guangzhou Laboratory (Bioland Laboratory, Guangzhou Regenerative Medicine and Health Guangdong Laboratory), Guangzhou, China
[10] Joint & Sports Medicine Surgery Division, Ruikang Hospital Affiliated to Guangxi University of Chinese Medicine, Nanning, Guangxi, China
* These authors contributed equally to this work.

Corresponding authors
Wenle Li, drlee0910@163.com
Zhe Feng, fengzhe2076@126.com

## ABSTRACT

**Background.** The occurrence of distant metastases (DM) limits the overall survival (OS) of patients with chondrosarcoma (CS). Early diagnosis and treatment of CS remains a great challenge in clinical practice. The aim of this study was to investigate metastatic factors and develop a risk stratification model for clinicians' decision-making.

**Methods.** Six machine learning (ML) algorithms, including logistic regression (LR), plain Bayesian classifier (NBC), decision tree (DT), random forest (RF), gradient boosting machine (GBM) and extreme gradient boosting (XGBoost). A 10-fold cross-validation was performed for each model separately, multicenter data was used as external validation, and the best (highest AUC) model was selected to build the network calculator.

**Results.** A total of 1,385 patients met the inclusion criteria, including 82 (5.9%) patients with metastatic CS. Multivariate logistic regression analysis showed that the risk of DM was significantly higher in patients with higher pathologic grades, T-stage, N-stage, and non-left primary lesions, as well as those who did not receive surgery and chemotherapy. The AUC of the six ML algorithms for predicting DM ranged from 0.911–0.985, with the extreme gradient enhancement algorithm (XGBoost) having the highest AUC. Therefore, we used the XGB model and uploaded the results to an online risk calculator for estimating DM risk.

**Conclusions**. In this study, combined with adequate SEER case database and external validation with data from multicenter institutions in different geographic regions, we confirmed that CS, T, N, laterality, and grading of surgery and chemotherapy were independent risk factors for DM. Based on the easily available clinical risk factors, machine learning algorithms built the XGB model that predicts the best outcome for DM. An online risk calculator helps simplify the patient assessment process and provides decision guidance for precision medicine and long-term cancer surveillance, which contributes to the interpretability of the model.

# INTRODUCTION

Chondrosarcoma (CS) is a rare malignant tumor that originates in cartilage tissue. Treatment usually includes surgery and radiation therapy (*Li et al., 2022*). Symptoms may include pain, swelling, and movement disorders. CS has an annual incidence of five per million, accounting for 30% of all malignant bone tumors, second only to osteosarcoma (*Biermann et al., 2017*; *Giuffrida et al., 2009*; *Song et al., 2018*). Studies have reported that the incidence of CS has been steadily increasing over the past half century, placing a tremendous burden on patients and their families (*Anfinsen et al., 2011*; *Whelan et al., 2012*). In the face of such an important malignant disease, the emergence of distant metastases (DM) has overshadowed the tragedy of CS. Identifying the risk of distal metastasis in patients with chondrosarcoma is very important as it affects the patient's treatment plan and prognosis (*Song et al., 2018*; *Thorkildsen et al., 2019*; *Thorkildsen et al., 2020*; *Tsuda et al., 2019*). Previous studies have concluded that the combination of surgery and chemotherapy is a reliable treatment option to improve patient prognosis (*Chen et al., 2017*; *Duffaud et al., 2019*; *Laitinen et al., 2019*), and precision medicine is an approach to formulate treatments in an individualized manner, which is based on the patient's characteristics, for example, a treatment plan that is quantitatively determined based on the risk of metastasis in chondrosarcoma. This can help improve treatment outcomes and reduce treatment side effects as well as personalization of cases. It has been shown that for chondrosarcoma, stratified identification of chondrosarcoma metastatic risk can lead to better treatment planning and improved prognosis. Therefore, precision medicine is very important for the treatment of chondrosarcoma patients and poses a great challenge to clinicians.

In previous studies, machine learning has pioneered novel and effective analytical methods for the medical field, especially those related to oncology (*Ngiam & Khor, 2019*; *Yamazawa et al., 2022*). Nevertheless, there are no reports about the application of machine learning in the evaluation of the metastases of CS. However, the current applications of machine learning in the field of chondrosarcoma are still relatively few and have yet to realize its full potential. Although studies have confirmed the feasibility and potential

benefits of machine learning in diagnosis and treatment, large-scale experiments and validation for clinical applications are still lacking, and data acquisition and data quality issues also need to be addressed for better application and dissemination in clinical practice (*Yue et al., 2022*). There are no reports on the application of machine learning to CS metastasis assessment, and only the studies by *Bongers et al. (2020)*, *Bongers et al. (2019)* and *Thio et al. (2018)* mainly focused on the analysis of the overall survival (OS) of CS using machine learning.

Given the low incidence of CS, we obtained sufficient cases from the Surveillance, Epidemiology, and End Results (SEER), and performed external validation based on data from multiple academic institutions.

## MATERIALS AND METHODS

### Data collection

The training group of this study included patients with CS diagnosed between 2010-2016. Cases in SEER* statistical software version 8.3.6 constituted the training group. The inclusion criteria were as follows: (1) patients whose primary tumor was only diagnosed as chondrosarcoma; (2) the histological morphology code was recorded as 9180 according to the third edition of the International Taxonomy of Oncology (ICDO-3); (3) patients whose survival status were available; (4) patients whose primary site was limited to bone and metastasis information was available.

External validation group included 104 patients from three Chinese academic medical centers. Data collection at each institution was completed by two researchers. One collected clinical and diagnostic data for each patient from the hospital's electronic records, and the other reviewed the data. The validation group consisted of 104 patients from three academic medical centers in China who were matched to the training group cohort on the basis of clinical characteristics (age, gender, treatment modality, pathology and clinical staging).

In order to assess the metastatic status of a patient, we use imaging tests such as CT and MRI scans to determine the presence of suspicious lesions. A suspicious lesion is identified if it has significant contrast enhancement on CT or MRI images. These findings were confirmed by a radiologist who did not maintain confidentiality of the patient's medical history.

The primary purpose of surgery was to eradicate CS, and we adhered to the surgical principle of extensive marginal excision, excluding palliative surgery for pain relief. The main chemotherapy drugs were gemcitabine, dacarbazine, trabectedin and taxanes.

In finalizing the data, we removed patients with missing values. In addition, the data collected from the electronic records were double-checked by the researchers and reviewed by two researchers to minimize any potential omissions, spelling errors, and inaccuracies and outliers.

Follow-up of patients every three months was recommended. Patients with suspicious metastases at follow-up had their metastatic status identified by systematic physical examination, laboratory tests, imaging and biopsy. This study was approved by the Ethics

Committee of Xianyang Central Hospital (Ethics number: 20210020). This study was retrospective, no medical intervention was performed on patients, and written informed consent was waived. This study complies with the Declaration of Helsinki and relevant ethical principles and laws and regulations will be strictly observed.

## Statistics analysis

A baseline table (Table 1) was drawn based on the presence or absence of distal metastases in patients with chondrosarcoma, which were obtained from the SEER database and by reviewing the patients' medical records, and included information on the patients' demographic characteristics, tumor characteristics, surgical procedures, chemotherapy, and follow-up. These variables were selected for analysis based on their potential association with chondrosarcoma and distant metastases. For example, variables such as age, gender, tumor size, histologic grade, and site of origin were considered important clinical characteristics of chondrosarcoma that may predict distant metastasis. Other variables, such as the presence of surgery, radiotherapy, and chemotherapy, as well as follow-up information, also provided valuable insights into the treatment of chondrosarcoma.

Logistic regression analysis was used to verify the relationships between distant metastasis and clinicopathological variables. According to the stepwise logistic regression process, variables ($P < 0.05$) in the univariate logistic regression analysis were included in the multivariate logistic. Risk factors confirmed by multivariate logistics with statistically significance were taken as independent risk factors of DM and finally included in the prediction model. The multivariate logistic regression process allowed us to identify the most important predictors of distant metastasis while controlling for confounders and potential bias. Predictor variables were ultimately selected based on their statistical significance and contribution to the overall predictive power of the model.

The statistical analysis in this study, including generation of baseline tables and correlation heat maps were carried out on R software (version 4.05). $P$ value $< 0.05$ was statistical significance.

## Machine learning algorithms

Compared with traditional statistical algorithms, machine learning algorithms provide greater credibility for clinical research of oncology. In this study, six ML algorithms were developed: logistic regression (LR), naive Bayesian classifier (NBC), decision tree (DT), random forest (RF), gradient boosting machine (GBM), and extreme gradient boosting (XGBoost). During the training, we adjusted the ML model to avoid overfitting. To compare the performance of ML algorithms, we used ROC curves to analyze the results. The closer the AUC is to 1, the better the performance of the classification model was and the stronger the predictive power was. Results from training group were cross-validated by 10-fold, and ML was further trained to predict the risk of DM *via* Python (version 3.8). The average AUC was used to evaluate the predictive power of each ML classifier. Subsequently, an online risk calculator was developed based on the best classifier, which can optimize the clinician's decision-making of DM risk by personalized predictions generated by typing in patient data.
**Table 1  Demographics and baseline characteristics of all patients.**

| Variables | Level | Overall (N = 1,385) | SEER data (N = 1,281) | Multicenter data (N = 104) | p |
|---|---|---|---|---|---|
| Race (%) | Black | 96 (6.9) | 96 (7.5) | 0 (0.0) | <0.001 |
| | Other | 181 (13.1) | 77 (6.0) | 104 (100.0) | |
| | White | 1,108 (80.0) | 1,108 (86.5) | 0 (0.0) | |
| Age (mean (SD)) | NA | 53.11 (17.93) | 53.40 (18.14) | 49.61 (14.63) | 0.038 |
| Sex (%) | Female | 605 (43.7) | 567 (44.3) | 38 (36.5) | 0.154 |
| | Male | 780 (56.3) | 714 (55.7) | 66 (63.5) | |
| Primary Site (%) | Axis bone | 730 (52.7) | 672 (52.5) | 58 (55.8) | 0.397 |
| | Bone of limb | 578 (41.7) | 540 (42.2) | 38 (36.5) | |
| | other | 77 (5.6) | 69 (5.4) | 8 (7.7) | |
| Laterality (%) | | 222 (16.0) | 201 (15.7) | 21 (20.2) | 0.567 |
| | left | 534 (38.6) | 494 (38.6) | 40 (38.5) | |
| | Not a paired site | 95 (6.9) | 90 (7.0) | 5 (4.8) | |
| | right | 534 (38.6) | 496 (38.7) | 38 (36.5) | |
| Grade (%) | Moderately differentiated | 555 (40.1) | 517 (40.4) | 38 (36.5) | 0.329 |
| | Poorly differentiated | 138 (10.0) | 125 (9.8) | 13 (12.5) | |
| | Undifferentiated; anaplastic | 40 (2.9) | 39 (3.0) | 1 (1.0) | |
| | Unknown | 177 (12.8) | 167 (13.0) | 10 (9.6) | |
| | Well differentiated | 475 (34.3) | 433 (33.8) | 42 (40.4) | |
| T (%) | T1 | 760 (54.9) | 713 (55.7) | 47 (45.2) | 0.007[**] |
| | T2 | 424 (30.6) | 386 (30.1) | 38 (36.5) | |
| | T3 | 14 (1.0) | 10 (0.8) | 4 (3.8) | |
| | TX | 187 (13.5) | 172 (13.4) | 15 (14.4) | |
| N (%) | N0 | 1,320 (95.3) | 1,229 (95.9) | 91 (87.5) | <0.001[***] |
| | N1 | 19 (1.4) | 10 (0.8) | 9 (8.7) | |
| | NX | 46 (3.3) | 42 (3.3) | 4 (3.8) | |
| M (%) | M0 | 1,303 (94.1) | 1,210 (94.5) | 93 (89.4) | 0.061 |
| | M1 | 82 (5.9) | 71 (5.5) | 11 (10.6) | |
| surgery (%) | No | 194 (14.0) | 175 (13.7) | 19 (18.3) | 0.248 |
| | Yes | 1,191 (86.0) | 1,106 (86.3) | 85 (81.7) | |
| Lymph node dissection (%) | No | 1,299 (93.8) | 1,204 (94.0) | 95 (91.3) | 0.388 |
| | Yes | 86 (6.2) | 77 (6.0) | 9 (8.7) | |
| Radiation (%) | No | 1,241 (89.6) | 1,145 (89.4) | 96 (92.3) | 0.440 |
| | Yes | 144 (10.4) | 136 (10.6) | 8 (7.7) | |
| Chemotherapy (%) | No/Unknown | 1,320 (95.3) | 1,223 (95.5) | 97 (93.3) | 0.435 |
| | Yes | 65 (4.7) | 58 (4.5) | 7 (6.7) | |
| times (mean (SD)) | NA | 34.27 (24.13) | 34.35 (24.15) | 33.29 (24.03) | 0.667 |

**Notes.**
[*]At 0.05 significance level.
[**]At 0.01 significance level.
[***]At 0.001 significance level.
The variables listed as "unknown" in Table 1 refer to patient-specific codes in the SEER database.

## RESULTS

### Demographic baseline characteristics

A total of 1,385 patients were enrolled, of which 1,281 were from the SEER database as a training group for the model and 104 were from three academic medical institutions in China as an external validation group. The variables with statistically significant differences were: race ($p < 0.001$), age ($p = 0.038$), T stage ($p = 0.007$), N stage ($p < 0.001$). The ethnicity of the multicenter data was predominantly Chinese (recorded as "other" races, 100%), while the race of the SEER data from the United States was white (86.5%). Our data showed that domestic patients with chondrosarcoma were younger on average (49.61 *vs* 53.40), with more non-T1 patients (54.8% *vs* 44.3%) and more non-N0 patients (12.5% *vs* 4.1%) (Table 1).

There are 82 (5.9%) CS patients diagnosed with DM, thus, mean differences have been re-analyzed based on this DM group. Results indicated that significant differences were observed in age ($p = 0.001$), gender ($p = 0.005$), laterality ($p = 0.003$), pathological grade ($p < 0.001$), T ($p < 0.001$), N ($p < 0.001$), survival time ($p < 0.001$), surgery ($p < 0.001$), and chemotherapy ($p < 0.001$) (Table 2).

### Univariate and multivariable logistic regression

The ROC curves assessed the predictive ability of the six ML algorithm models in the validation group. The training group was validated using 10-fold cross-validation, and the results showed that the XGB model had an average AUC of 0.985, which significantly outperformed the other MLs (Fig. 1). The XGB model also occupied the largest area in the ROC of the external validation group, indicating that it was still the most efficient equation (Fig. 2), and thus the XGB model was ultimately selected as the predictive model.

The results of logistic regression analysis were shown in Table 3. Univariate logistic analysis was carried out for age ($p = 0.001$), gender ($p = 0.003$), grade ($p < 0.005$), Laterality ($p < 0.005$), T ($p < 0.001$), N ($p < 0.001$), surgery ($p < 0.001$) and chemotherapy drugs ($p < 0.001$) on DM. The odds ratio (OR) revealed the factors associated with DM. In multivariate logistic analysis, we found that patients with pathological grade, T-stage, N-stage and non-left primary lesions, as well as those who did not receive surgery and chemotherapy, had a significantly higher risk of DM at 0.05 significance level (Table 2).

### Clinical predictive model development

The relative importance of the identified risk factors was shown in Fig. 3. A general trend was seen in the evidence: although the importance of each variable varied slightly, surgery was still ranked first in three algorithms (LR, RF and GBM), chemotherapy ranked first in two algorithms (NBC and DT), and age ranked first in one algorithm (XGB). In contrast, laterality, T-stage, N-stage, and gender had little effect across multiple prediction models. The risk factors of the XGB model were listed as follows by correlation: age, surgery, chemotherapy, grade, laterality, T, gender, and N (Fig. 3). The association between risk factors was also shown in the correlation heat map, and the closer the yellows is, the greater the intensity is (Fig. 4). No significant association has been proven among these independent risk factors in this study.

**Table 2  Characteristics of chondrosarcoma with distant metastases and non-distant metastases.**

| Variables | Level | Overall (N = 1,385) | NDM (N = 1,303) | DM (N = 82) | p |
|---|---|---|---|---|---|
| category (%) | Multicenter data | 104 (7.5) | 93 (7.1) | 11 (13.4) | 0.061 |
| | SEER data | 1,281 (92.5) | 1,210 (92.9) | 71 (86.6) | |
| Race (%) | Black | 96 (6.9) | 90 (6.9) | 6 (7.3) | 0.523 |
| | Other | 181 (13.1) | 167 (12.8) | 14 (17.1) | |
| | White | 1,108 (80.0) | 1,046 (80.3) | 62 (75.6) | |
| Age (mean (SD)) | NA | 53.11 (17.93) | 52.70 (17.85) | 59.68 (17.89) | 0.001** |
| Sex (%) | Female | 605 (43.7) | 582 (44.7) | 23 (28.0) | 0.005* |
| | Male | 780 (56.3) | 721 (55.3) | 59 (72.0) | |
| Primary Site (%) | Axis bone | 730 (52.7) | 681 (52.3) | 49 (59.8) | 0.232 |
| | Bone of limb | 578 (41.7) | 551 (42.3) | 27 (32.9) | |
| | other | 77 (5.6) | 71 (5.4) | 6 (7.3) | |
| Laterality (%) | left | 534 (38.6) | 517 (39.7) | 17 (20.7) | 0.003** |
| | Not a paired site | 317 (22.9) | 291 (22.3) | 26 (31.7) | |
| | right | 534 (38.6) | 495 (38.0) | 39 (47.6) | |
| Grade (%) | Moderately differentiated | 555 (40.1) | 526 (40.4) | 29 (35.4) | <0.001*** |
| | Poorly differentiated | 138 (10.0) | 123 (9.4) | 15 (18.3) | |
| | Undifferentiated; anaplastic | 40 (2.9) | 33 (2.5) | 7 (8.5) | |
| | Unknown | 177 (12.8) | 152 (11.7) | 25 (30.5) | |
| | Well differentiated | 475 (34.3) | 469 (36.0) | 6 (7.3) | |
| T (%) | T1 | 760 (54.9) | 747 (57.3) | 13 (15.9) | <0.001*** |
| | T2 | 424 (30.6) | 382 (29.3) | 42 (51.2) | |
| | T3 | 14 (1.0) | 10 (0.8) | 4 (4.9) | |
| | TX | 187 (13.5) | 164 (12.6) | 23 (28.0) | |
| N (%) | N0 | 1,320 (95.3) | 1,256 (96.4) | 64 (78.0) | <0.001*** |
| | N1 | 19 (1.4) | 12 (0.9) | 7 (8.5) | |
| | NX | 46 (3.3) | 35 (2.7) | 11 (13.4) | |
| Times (mean (SD)) | NA | 34.27 (24.13) | 35.47 (23.94) | 15.20 (18.58) | <0.001*** |
| Surgery (%) | No | 194 (14.0) | 145 (11.1) | 49 (59.8) | <0.001*** |
| | Yes | 1191 (86.0) | 1158 (88.9) | 33 (40.2) | |
| Lymph node dissection (%) | No | 1,299 (93.8) | 1,222 (93.8) | 77 (93.9) | 1.000 |
| | Yes | 86 (6.2) | 81 (6.2) | 5 (6.1) | |
| Radiation (%) | No | 1,241 (89.6) | 1,165 (89.4) | 76 (92.7) | 0.450 |
| | Yes | 144 (10.4) | 138 (10.6) | 6 (7.3) | |
| Chemotherapy (%) | No/Unknown | 1,320 (95.3) | 1,265 (97.1) | 55 (67.1) | <0.001*** |
| | Yes | 65 (4.7) | 38 (2.9) | 27 (32.9) | |

**Notes.**
DM, distance metastases; NDM, non-distance metastases.
*At 0.05 significance level.
**At 0.01 significance level.
***At 0.001 significance level.

## Web calculator design

We provided users with a web-based calculator, a digital tool based on the best predictive performance modle (XGB model) (https://share.streamlit.io/liuwencai2/chs_m/main/chs_m.py) (Fig. 5).

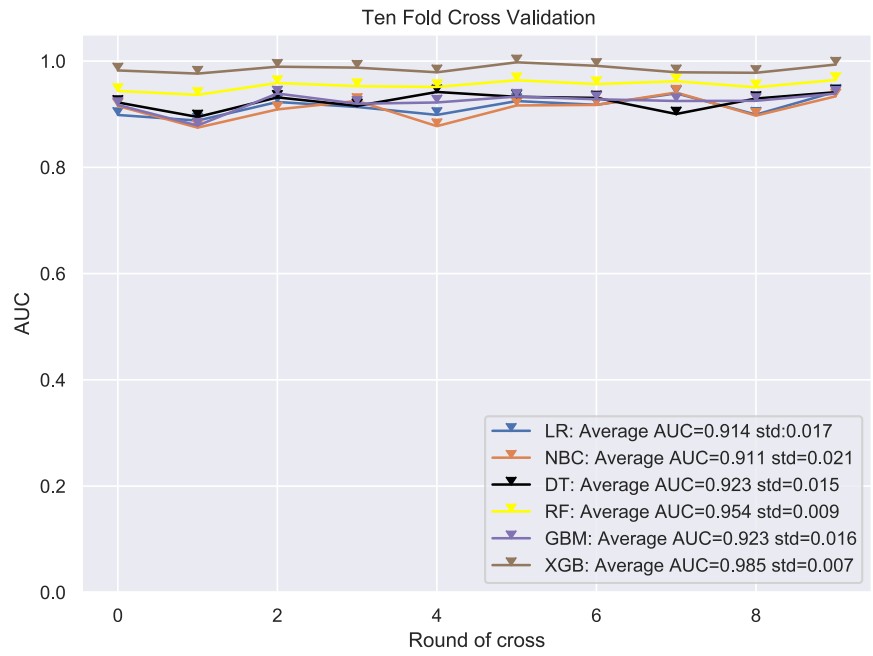

**Figure 1  10-fold cross validation test.** LR, logistic regression; MLP, multilayer perceptron; DT, decision tree; RF, random forest; GBM, gradient boosting machine; XGB, extreme gradient boosting.

## DISCUSSION

Accurate diagnosis of metastatic status is a critical component of oncology management, and a large amount of scholars are dedicated to promoting this field (*Chan et al., 2015*; *Sheen et al., 2019*; *Song et al., 2018*; *Zhang et al., 2021*). CS poses a serious threat to patients with OS, especially for patients with DM, where treatment is severely limited and prognosis deteriorates dramatically (*Thorkildsen et al., 2019*; *Thorkildsen et al., 2020*; *Tsuda et al., 2019*). In this regard, prominent contributions have been made by *Song et al. (2018)* to facilitate the risk stratification of patients with DM, pointing out that this challenge must be overcome on the way to precision medicine. In fact, machine learning algorithms have been widely accepted as a disease investigation tool from which patients can benefit, and these novel mathematical tools could demonstrate superior performance to traditional statistical methods by inputting sufficient amounts of clinical data (*Gao et al., 2023*; *Li et al., 2021*; *Pereira et al., 2021*). However, the extremely low incidence of CS has prevented most investigators from exploring the clinical characteristics of patients thoroughly. The SEER database provides a promising solution to this dilemma. Furthermore, compared with previous reports, the multicenter validation datasets within China in this study is more convincing.

Our study found that high histologic grades was a key factor in DM and revealed statistical differences between the two populations. Pathological grade was a recognized prognostic factor for CS. Grade I sarcomas were well differentiated, while grade II and

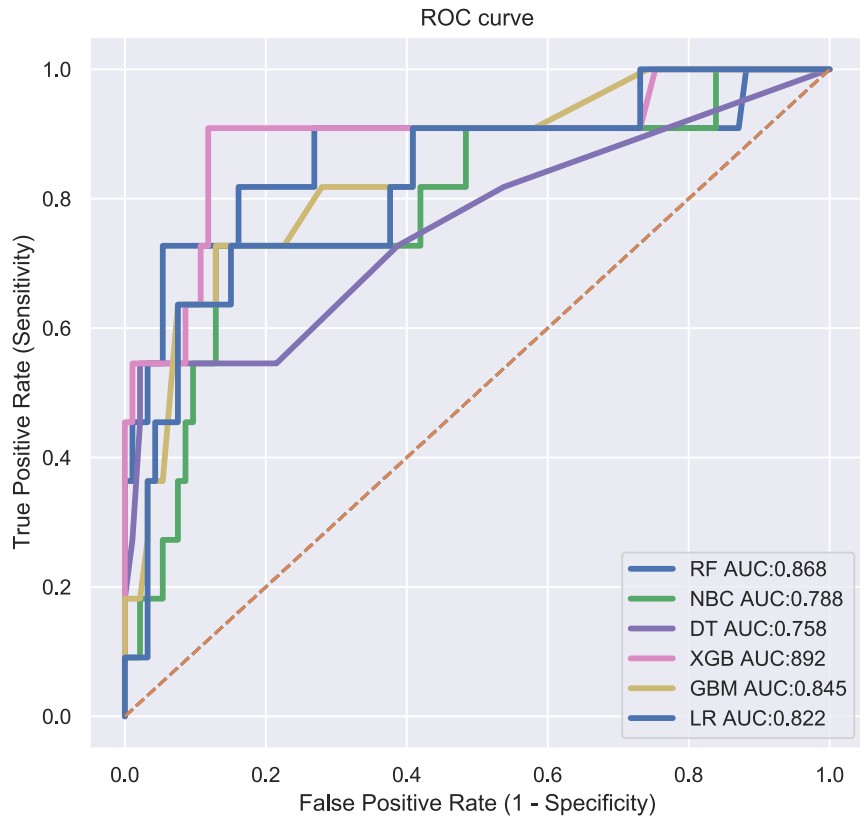

**Figure 2  ROC curves evaluating the predictive power of the six ML algorithm models.** LR, logistic regression; MLP, multilayer perceptron; DT, decision tree; RF, random forest; GBM, gradient boosting machine; XGB, extreme gradient boosting.

III sarcoma were malignant (*Thorkildsen et al., 2019*; *Thorkildsen et al., 2020*). Grade II tumors were distinguished by nuclear enlargement, hyperchromatism, and changes in size and shape (*Weinschenk, Wang & Lewis, 2021*). Grade III chondrosarcomas typically had more prominent osteolytic changes and tend to have cortical breakthroughs in soft tissue extension. These malignant cells showed marked nuclear heterogeneity, pleomorphism, and mitosis. The local recurrence rate of CS with poor histologic grade was increased, with metastases occurring in more than 50%, and the most common site of involvement was in the lung (*Weinschenk, Wang & Lewis, 2021*). It was noteworthy that even in patients underwent surgery, 11% of tumors were of higher grade than originally diagnosed once recurrence or metastasis occurred (*Pritchard et al., 1980*).

The American Joint Committee on Cancer (AJCC) Staging System Version 8 recommends the use of 8 cm as the cut-off value for risk stratification (*Amin et al., 2017*). Although the association between tumor size and patient prognosis has not been well established, *Roos et al. (2016)* and *Wang et al. (2019)* found that patients with smaller tumor tended to have longer survival. Nevertheless, *Kamal et al. (2015)* and *Bindiganavile et al. (2015)* rejected the predictive role of tumor size in CS. We concluded that high-grade

**Table 3** Univariate and multifactorial logistic regression analysis of risk factors for metastases in patients with chondrosarcoma.

| Variables | Univariate | | | Multivariate | | |
|---|---|---|---|---|---|---|
| | OR | 95% CI | p value | OR | 95% CI | p value |
| Age (years) | 1.023 | 1.009–1.036 | 0.001 | 1.005 | 0.983–1.027 | 0.097 |
| Race | | | | | | |
| White | Ref | | Ref | Ref | | Ref |
| Black | 1.106 | 0.466–2.625 | 0.820 | / | | / |
| Other | 1.391 | 0.762–2.538 | 0.283 | / | | / |
| Sex | | | | | | |
| Male | Ref | | Ref | Ref | | Ref |
| Female | 0.474 | 0.290–0.776 | 0.003 | 0.589 | 0.322–1.076 | 0.085 |
| Primary site | | | | | | |
| Limb bones | Ref | | Ref | Ref | | Ref |
| Axis of a bone | 1.501 | 0.927–2.428 | 0.098 | / | | / |
| other | 1.725 | 0.688–4.321 | 0.245 | / | | / |
| Grade | | | | | | |
| I | Ref | | Ref | Ref | | Ref |
| II | 3.701 | 1.606–8.528 | 0.002 | 2.112 | 0.810–5.510 | 0.126 |
| III | 8.171 | 3.260–20.479 | 0.000 | 3.224 | 1.075–9.667 | 0.037[*] |
| IV | 14.212 | 4.705–42.932 | 0.000 | 1.987 | 0.502–7.869 | 0.328 |
| V | 11.020 | 4.673–25.985 | 0.000 | 3.037 | 1.111–8.307 | 0.030[*] |
| Laterality | | | | | | |
| Left | Ref | | Ref | Ref | | Ref |
| Right | 2.396 | 1.338–4.291 | 0.003 | 2.281 | 1.103–4.715 | 0.026[*] |
| Other | 2.831 | 1.518–5.283 | 0.001 | 2.373 | 1.057–5.328 | 0.036[*] |
| T | | | | | | |
| T1 | Ref | | Ref | Ref | | Ref |
| T2 | 6.485 | 3.445–12.207 | 0.000 | 3.445 | 1.680–7.067 | 0.001[**] |
| T3 | 22.985 | 6.374–82.885 | 0.000 | 18.774 | 4.130–85.339 | 0.000[***] |
| TX | 8.059 | 3.999–16.241 | 0.000 | 2.578 | 1.079–6.160 | 0.033[*] |
| N | | | | | | |
| N0 | Ref | | Ref | Ref | | Ref |
| N1 | 11.263 | 4.291–29.560 | 0.000 | 4.289 | 1.164–15.805 | 0.029[*] |
| NX | 5.068 | 2.948–12.490 | 0.000 | 3.348 | 1.271–8.820 | 0.014[*] |
| Surgery | | | | | | |
| No | Ref | | Ref | Ref | | Ref |
| Yes | 0.087 | 0.054–0.139 | 0.000 | 0.208 | 0.111–0.389 | 0.000[***] |
| Lymph node dissection | | | | | | |
| No | Ref | | Ref | Ref | | Ref |
| Yes | 0.966 | 0.381–2.453 | 0.942 | | | |
| Radiation | | | | | | |
| No | Ref | | Ref | Ref | | Ref |
| Yes | 0.777 | 0.351–1.719 | 0.533 | / | | / |

**Table 3** (*continued*)

| Variables | Univariate | | | Multivariate | | |
|---|---|---|---|---|---|---|
| | **OR** | **95% CI** | **p value** | **OR** | **95% CI** | **p value** |
| Chemotherapy | | | | | | |
| No | Ref | | Ref | Ref | | Ref |
| Yes | 16.934 | 9.694–29.581 | 0.000 | 11.623 | 571–5.24.252 | 0.000[***] |

**Notes.**

Grade I, Well-differentiated; Grade II, Moderately differentiated; Grade III, Poorly differentiated; Grade IV, Undifferentiated anaplastic; Grade V, Unknown.

[*]At 0.05 significance level.

[**]At 0.01 significance level.

[***]At 0.001 significance level.

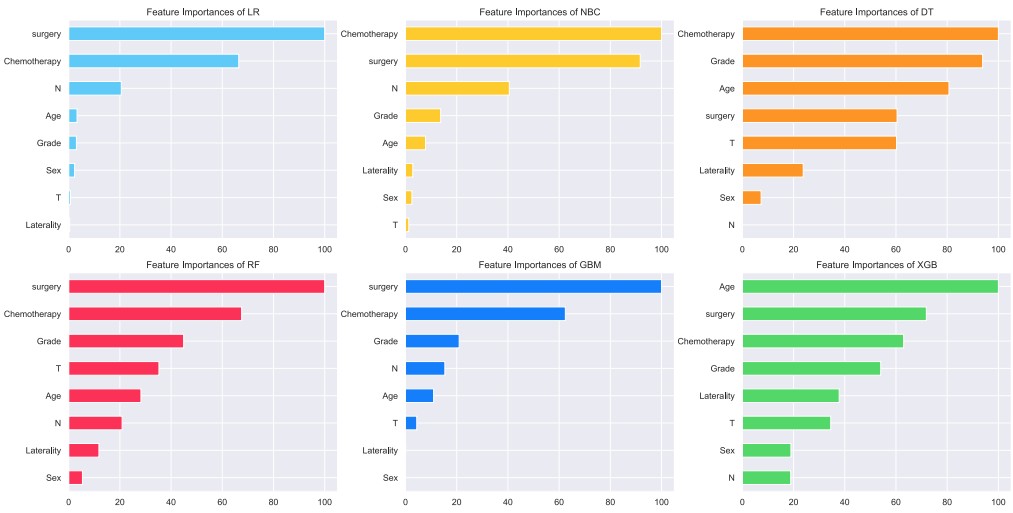

**Figure 3** **Patients clinical features importance of six ML algorithm models.**

T-stage suggested a high risk of DM. This may be linked to the tendency of larger tumors to break through the borders and thus invade the peripheral vessels. In addition, N-stage, a classifier indicating lymphatic status of the tumor, was identified as an independent risk factor for DM. Although lymphatic metastasis is considered to be a threat to patient survival, we believe that inclusion of the independent validation group in the study design ensured convincing conclusions, given the large sample size reported from the SEER database (*Basile et al., 2020*; *Giuffrida et al., 2009*; *Wan et al., 2019*).

Negative margin surgery resulted in fewer DM and improved survival rates (*Weinschenk, Wang & Lewis, 2021*). However, currently unclear definitions have prevented studies from agreeing on the evaluation of this risk factor. In the reports of *Streitbuerger et al. (2012)* and *Laitinen et al. (2019)* patients did not receive the same incidence of margin-negative surgery (76% *vs* 49%), but achieved the same survival. In the view of the above, we recommend Enneking's classification system, which requires an acceptable negative edge of the normal tissue cuff when removing the involved portion of the bone. Ensuring consistent

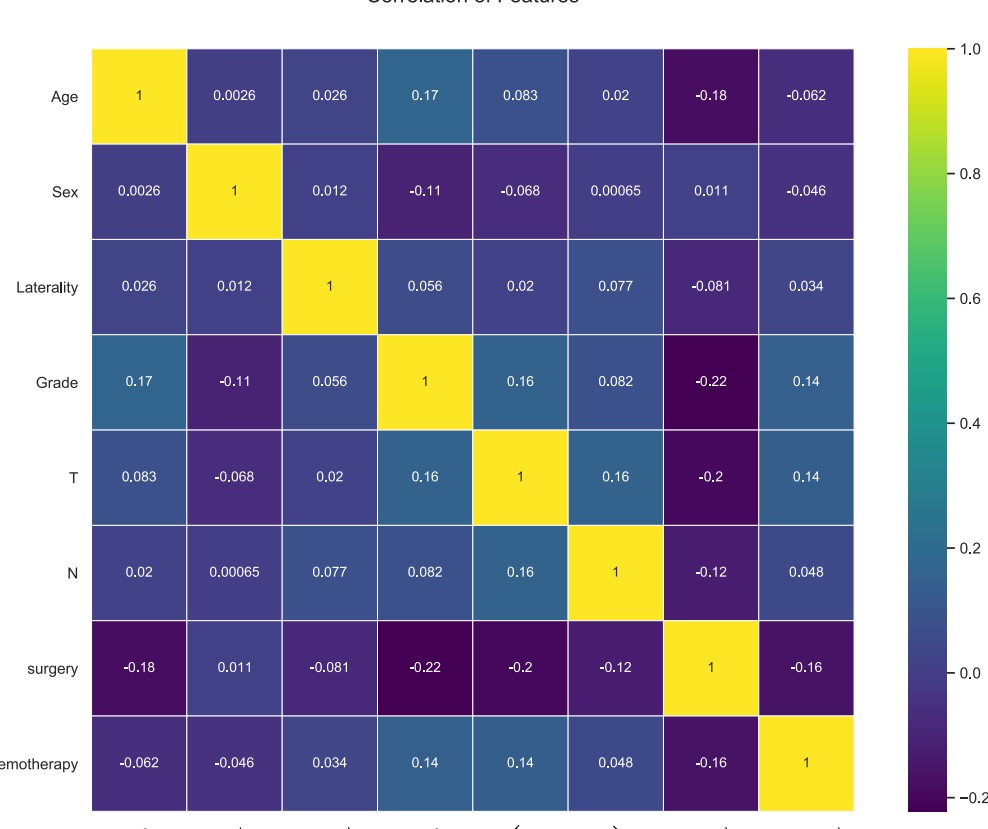

**Figure 4** **Web calculator for the risk of distal metastasis of chondrosarcoma.**

variable measurement criteria is the first step in achieving predictive validity. Meanwhile, preoperative neoadjuvant therapy is expected to effectively improve the microenvironment of the sarcoma, which is considered to be the intention of radical surgery (*Casali et al., 2018*; *Radaelli et al., 2014*). *Frustaci et al. (2001)* and *Gronchi et al. (2012)* found that the chemotherapy regimen of epirubicin, ifosfamide and G-CSF can help control patients' DM. Interestingly, our findings are consistent with published studies. Preoperative radiotherapy without neoadjuvant therapy does not seem to help control the local recurrence and metastasis of CS (*Radaelli et al., 2014*; *Strander, Turesson & Cavallin-Ståhl, 2003*).

The XGB model (best predictive performance) suggested that age played an extraordinary role in DM of CS. Compared with data from SEER database, the average age of patients in this study was younger, which may be due to lack of rational management, resulting in higher tumor-specific death and shorter tumor-free survival (*Wang et al., 2020*). In national Norwegian studies, older patients were found to have larger tumors and a higher risk of DM (*Thorkildsen et al., 2019*; *Thorkildsen et al., 2020*). This may be because longer exposure to tumors leads to an increased likelihood that malignant cells will spread hematologically.

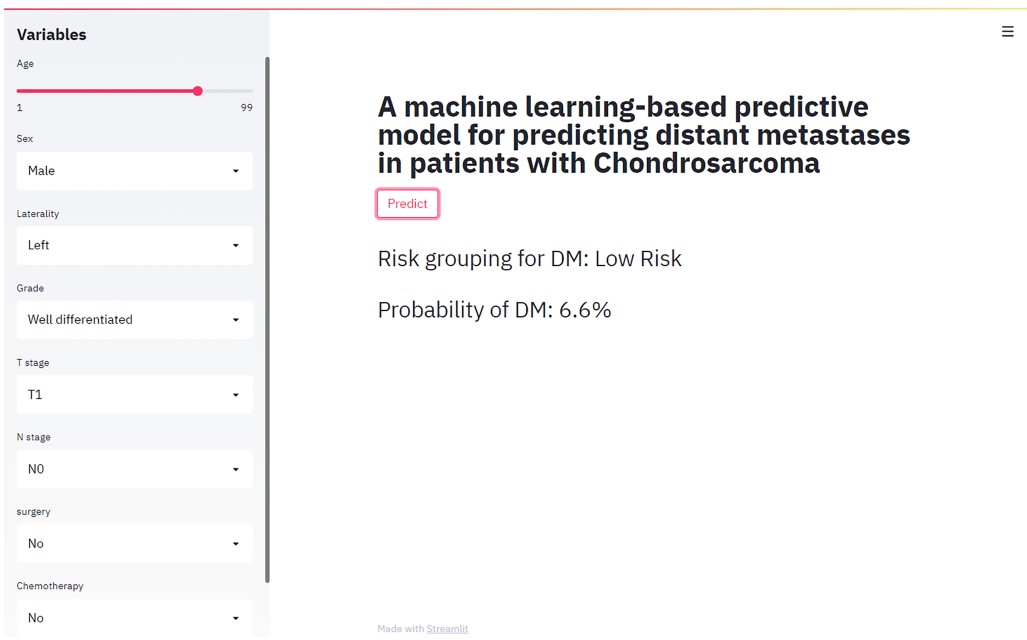

**Figure 5** **Heat map of the correlation of patients' clinical features.**

Additionally, advanced age consistently predicted shorter OS in survival analyses after local recurrence or DM (*Fromm et al., 2018*; *Laitinen et al., 2019*; *Song et al., 2018*).

Traditional protocols for tumor surveillance and personalized treatment decisions include clinical presentation and symptoms, molecular markers, periodic follow-up, and an experienced physician team. These methods can be used as adjuncts, but may lack precision and reproducibility compared to accurate stratification, and are costly. Machine learning algorithms can be used as an effective tool to predict the risk of chondrosarcoma metastasis. With machine learning algorithms, we can better understand the risk factors associated with distal metastasis in order to predict and treat patients as early as possible, thereby improving the effectiveness of treatment and patient survival. Second, the online risk calculator in this study provides an easy-to-use tool to help clinicians quickly measure a patient's risk of metastasis and better plan diagnosis and treatment. In addition, this study highlights the importance of chemotherapy and surgery, which is informative for clinicians to choose the best treatment options.

In summary, the use of machine learning algorithms helped determine a formula that accurately classified CS patients into high-risk and low-risk groups according to the probability of DM. However, the study has some limitations. First, it is a retrospective study, which needs to be further verified by prospective research. Second, despite we attempted to define treatment options, the SEER database does not provide details of surgery and chemotherapy, suggesting that this risk factor requires further subgroup analysis to determine its role in DM. In addition, the predictions of the algorithm may not be accurate enough due to flaws in the algorithm or the quality of the sample data, while the external validation group only contains cohort data from China, so further

validation is still needed for use in other regions. Machine-learning algorithms cannot completely replace the practical experience and expertise of clinicians, and algorithms need to be more interpretable so that physicians better understand the factors that identify and predict distal metastasis in chondrosarcoma. Finally, the included clinical data may lack characterization, and the unique biomarkers and radiological findings of CS may be potentially associated with DM (*Jeong & Kim, 2018*; *Miwa et al., 2021*; *Nazarizadeh et al., 2021*). To improve the accuracy and generalization of this model by integrating multi-modal and multi-dimensional data is highly recommended in future studies.

## CONCLUSIONS

We developed an online machine learning computational tool to predict the risk of distant metastasis in chondrosarcoma patients using generally available clinical data. A multicenter external validation group showed the model to have a little clinical value and could help clinicians to perform further screening of high-risk patients. In conclusion, rigorous follow-up was strongly recommended for patients with advanced pathological grade, advanced T-stage and N-stage, laterality and failure to undergo surgery and chemotherapy, due to the high risk of metastasis.

### Funding

This work was supported by grants from National Natural Science Foundation of China (82260944), the Key research and development programs of Guangxi, China (2021AB09011) and the Innovation Project of Guangxi Graduate Education of GXUCM, (YCBXJ2023006). The Guangxi Science and Technology Major Program, Grant No. GUIKEAA23023035 supported the APC of this article. The funders had no role in study design, data collection and analysis, decision to publish, or preparation of the manuscript.

### Grant Disclosures

The following grant information was disclosed by the authors:
National Natural Science Foundation of China: 82260944.
Key research and development programs of Guangxi, China: 2021AB09011.
Innovation Project of Guangxi Graduate Education of GXUCM: YCBXJ2023006.
Guangxi Science and Technology Major Program: GUIKEAA23023035.

### Competing Interests

The authors declare there are no competing interests.

### Author Contributions

- Jihu Wei performed the experiments, analyzed the data, authored or reviewed drafts of the article, and approved the final draft.
- Shijin Lu performed the experiments, analyzed the data, authored or reviewed drafts of the article, and approved the final draft.

- Wencai Liu performed the experiments, analyzed the data, authored or reviewed drafts of the article, and approved the final draft.
- He Liu analyzed the data, authored or reviewed drafts of the article, and approved the final draft.
- Lin Feng analyzed the data, prepared figures and/or tables, and approved the final draft.
- Yizi Tao analyzed the data, prepared figures and/or tables, and approved the final draft.
- Zhanglin Pu analyzed the data, prepared figures and/or tables, and approved the final draft.
- Qiang Liu analyzed the data, prepared figures and/or tables, and approved the final draft.
- Zhaohui Hu analyzed the data, prepared figures and/or tables, and approved the final draft.
- Haosheng Wang analyzed the data, prepared figures and/or tables, and approved the final draft.
- Wenle Li conceived and designed the experiments, prepared figures and/or tables, authored or reviewed drafts of the article, and approved the final draft.
- Wei Kang conceived and designed the experiments, prepared figures and/or tables, authored or reviewed drafts of the article, and approved the final draft.
- Chengliang Yin conceived and designed the experiments, prepared figures and/or tables, authored or reviewed drafts of the article, and approved the final draft.
- Zhe Feng conceived and designed the experiments, prepared figures and/or tables, authored or reviewed drafts of the article, and approved the final draft.

## Data Availability

The data from the SEER database (https://seer.cancer.gov/) and external validation is available in the Supplementary File.

## Supplemental Information

Supplemental information for this article can be found online at http://dx.doi.org/10.7717/peerj.16485#supplemental-information.

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
