# Peer review of "A machine learning-based model for clinical prediction of distal metastasis in chondrosarcoma: a multicenter, retrospective study"

_PeerJ, doi:10.7717/peerj.16485_

## Round 0.1 · original submission · Major Revisions

Some parts of this article need major revisions to improve the quality and clarity of the content. My suggestions are as follows

1. Clarify the research question and objectives. The introduction does not clearly state the research question and objectives. There is a need for a clear and concise statement of the research questions and objectives that the study aims to answer.

2. Improve the methodology section: The description of the methodology used in this study is not sufficient and should be revised. The rationale for the selection of the particular machine learning algorithm should be provided. In addition, the criteria for selecting patients for the training and validation groups need to be made more explicit.

3. Provide more descriptive statistics: The article should provide more descriptive statistics for detailed analysis of the data. The authors should provide means and standard deviations for continuous variables and percentages for categorical variables. These details will help the reader to better understand the characteristics of the study sample and enhance the interpretation of the results.

4. Strengthen the results section. The results section should be reorganized and presented in a more coherent and logical manner. The findings should be organized into subsections according to the objectives of the study. The author should focus on the main results and avoid extraneous information.

Reviewer 1 ·

Basic reporting

Thanks for the invitation to review the article, which was modeled using the SEER database and validated with external data, using a variety of machine learning algorithms, with a little bit of innovation and clinical value, but in still has some problems that need to be revised:
1. the introduction needs to focus more on the research gaps and limitations in current clinical practice to justify the study.
2. more information is needed on how they collected and assessed the quality of the data, including missing values, outliers, and potential bias.

Experimental design

3. tables 1 and 2 need further explanation of how the variables were collected. There also needs to be an explanation of why some of these variables were included in the analysis and how they relate to the objectives of the study.
4. the authors need to provide more information about the statistical methods used and their assumptions, including how missing data were handled, model assumptions, and variable selection criteria.
5. it is not clear why the authors chose logistic regression analysis as the primary method for assessing DM, given that machine learning methods were the focus of the study. They need to clarify the rationale for this choice and compare the results of logistic regression with those of machine learning algorithms.

Validity of the findings

6. the authors need to provide more details about the validation process, including how the dataset was divided, the performance metrics used for evaluation, and comparisons with other similar studies.
7. the discussion section should focus more on the implications of the findings for clinical practice, including the potential benefits and limitations of using machine learning algorithms to diagnose and treat patients with CS.
8 The authors should provide more detailed explanations of the terminology and medical jargon used in the manuscript to improve readability and accessibility for lay readers.

Additional comments

9. The conclusion section should summarize the main findings of the study, limitations, and implications for future research.

·

Basic reporting

The author's writing level is proficient and I believe it meets the requirements of the journal.

Experimental design

The design of the study is reasonable and credible, in accordance with the general research process of the relevant research.

Validity of the findings

The findings of this study are reliable, and the relevant research results have certain clinical reference significance.

Additional comments

1. the first sentence should include a brief explanation of chondrosarcoma (CS) for readers unfamiliar with the term.
2. include a full source citation for the study mentioned in the introduction.
3. discuss how the increased incidence of CS affects patients and their families.
4. explain the criteria used to identify patients at risk for DM and provide evidence to support the claim that accurate stratification is precision medicine.
5. provide more information about the potential advantages and disadvantages of using machine learning to assess CS metastasis.
6. discuss the generalizability of the study results given the limited number of cases in the SEER database, the value of using it in other centers
7. clarify the process used to select patients for inclusion in the study.
8. explain the specific role of researchers in data collection in academic medical centers
9. discuss the potential limitations of including only patients from Chinese institutions in the external validation group.
10. provide more detailed information about the chemotherapeutic agents used in this study, including dosages and routes of administration.
11. expand on the strengths and weaknesses of the statistical analysis and discuss potential sources of bias.
12. clarify how the retrospective nature of the study affects its validity and generalizability.
13. discuss the potential impact of the findings on future research and the development of precision medicine.
14. provide recommendations for further research and improvement of research methods and analyses.

Reviewer 3 ·

Basic reporting

Good

Experimental design

Moderate

Validity of the findings

ok

Additional comments

This paper uses the data of the SEER database to construct a risk model for predicting remote transfer and uses an external database to verify the generalization performance of the model, and the overall method ↑ meets the construction specification of the predictive model, but there are still some problems to be solved:

The authors do not reflect in the introduction a more precise relationship between the study and existing research, and a clearer link needs to be established

Findings from previous studies relevant to the treatment of patients with high-risk DM in CS need to be summarized

Discuss potential alternatives to tumor surveillance and personalized treatment decisions in addition to accurate stratification.

Describe in more detail the process of acquiring and processing data from the SEER database and discuss any potential limitations or biases that the dataset may have.

Further clarification is needed on the rationale for the exclusion criteria used to select study patients. Provide more information about data collection methods, including any potential sources of bias or errors.

Additional discussion of potential limitations of possible external validity in validating the specific Chinese context of a group needs to be discussed. Include more detailed information about chemotherapy drugs, including any potential side effects or contraindications. Discuss further potential implications of research on patient care and decision-making beyond what is mentioned in the conclusions.

The author needs to provide more context and details about the logistic regression analysis, including the specific variables and calculations used.

Potential limitations or sources of bias in statistical analysis and how these may affect the results of the study need to be discussed. Explain in more detail the ethical implications of retrospective research.

---

## Round 0.2 · Minor Revisions

Please describe details about the matching process for selecting the external validation.

How were suspicious metastases identified and recognized? Are radiologists blinded to results?

Reviewer 1 ·

Basic reporting

Thank you for submitting a revised version of your article, I have reviewed the manuscript again and the article has been greatly improved but there are still some minor issues that need to be addressed before taking over:

1. how do researchers assess a patient's metastatic status? How were suspicious metastases identified and recognized?

Experimental design

2. in Table 1, several variables are listed as "unknown". Can you provide more information about why these variables are unknown and how this might affect the results of the analysis?

Validity of the findings

3. in the methods section it needs to be added how patient confidentiality and privacy were protected and how the study adhered to international standards of research ethics .

Additional comments

4. in the methods section, it will help to clarify the matching process used to select the external validation group. How were 104 patients from three academic medical centers in China matched to the training group?

·

Basic reporting

good

Experimental design

good

Validity of the findings

good

---

## Round 0.3 · accepted · Accept

Authors have made revisions according to all comments and meet reviewers' requirements. This is a multicenter study with big sample size, of which the evidence is relatively strong. Therefore, I think this paper can be accepted for publication.

Reviewer 1 ·

Basic reporting

This machine learning and chondrosarcoma related study has been revised to meet the required content for the relevant type of study, I have no further questions about it, it has met the publication requirements, and I recommend it to the editors for acceptance.

Experimental design

none

Validity of the findings

none

Additional comments

none

·

Basic reporting

The language of this paper complies with norms, the writing is fluent, and the citations are appropriate.

Experimental design

The design is reasonable, the content is complete, and a reliable conclusion can be obtained.

Validity of the findings

The conclusion is reliable and can bring some new inspiration and gains to readers.

Additional comments

The study has been reworked to generally respond to my concerns and further details added, and is now a relatively rigorous study, so I recommend the article for acceptance and publication.

Reviewer 3 ·

Basic reporting

Good

Experimental design

Good

Validity of the findings

Good

Additional comments

After the revision, the author's description of the experimental design and data processing is more detailed, has sufficiently described the overall details of the study, and the analysis of the model is more rigorous, which has met the needs of the article for publication, and I agree to receive this manuscript.